# Oral feeding with probiotic *Lactobacillus rhamnosus* attenuates cigarette smoke-induced COPD in C57Bl/6 mice: Relevance to inflammatory markers in human bronchial epithelial cells

J. L. Carvalho[1], M. Miranda[1], A. K. Fialho[1], H. Castro-Faria-Neto[2], E. Anatriello[1], A. C. Keller[3], F. Aimbire[1] *

**1** Department of Science and Technology, Federal University of São Paulo, São José dos Campos, São Paulo, Brazil, **2** Laboratory of Immunopharmacology, FioCruz, Rio de Janeiro, Brazil, **3** Department of Microbiology, Immunology and Parasitology, Federal University of São Paulo, São Paulo, São Paulo, Brazil

* flavio.aimbire@unifesp.br

**Data Availability Statement:** All relevant data are within the manuscript.

## Abstract

COPD is a prevalent lung disease with significant impacts on public health. Affected airways exhibit pulmonary neutrophilia and consequent secretion of pro-inflammatory cytokines and proteases, which result in lung emphysema. Probiotics act as nonspecific modulators of the innate immune system that improve several inflammatory responses. To investigate the effect of *Lactobacillus rhamnosus* (*Lr*) on cigarette smoke (CS)-induced COPD C57Bl/6 mice were treated with *Lr* during the week before COPD induction and three times/week until euthanasia. For *in vitro* assays, murine bronchial epithelial cells as well as human bronchial epithelial cells exposed to cigarette smoke extract during 24 hours were treated with Lr 1 hour before CSE addition. *Lr* treatment attenuated the inflammatory response both in the airways and lung parenchyma, reducing inflammatory cells infiltration and the production of pro-inflammatory cytokines and chemokines. Also, *Lr*-treated mice presented with lower metalloproteases in lung tissue and lung remodeling. In parallel to the reduction in the expression of TLR2, TLR4, TLR9, STAT3, and NF-κB in lung tissue, *Lr* increased the levels of IL-10 as well as SOCS3 and TIMP1/2, indicating the induction of an anti-inflammatory environment. Similarly, murine bronchial epithelial cells as well as human bronchial epithelial cells (BEAS) exposed to CSE produced pro-inflammatory cytokines and chemokines, which were inhibited by *Lr* treatment in association with the production of anti-inflammatory molecules. Moreover, the presence of *Lr* also modulated the expression of COPD-associated transcription found into BALF of COPD mice group, i.e., *Lr* downregulated expression of NF-κB and STAT3, and inversely upregulated increased expression of SOCS3. Thus, our findings indicate that *Lr* modulates the balance between pro- and anti-inflammatory cytokines in human bronchial epithelial cells upon CS exposure and it can be a useful tool to improve the lung inflammatory response associated with COPD.

**Funding:** - Flavio Aimbire (FA) This work was supported by Fundação de Amparo à Pesquisa do Estado de São Paulo (FAPESP) Grant: 2015/18455-0. The funder had no role in study design, data collection and analysis, decision to publish, or preparation of the manuscript.

**Competing interests:** The authors have declared that no competing interest exist.

## 1. Introduction

Although chronic obstructive pulmonary disease (COPD) is one of the major chronic health conditions in which disability and death rates are increasing worldwide, the development of new strategies to disease management remains underwhelming [1–3]. Although the intrinsic factors that contribute to COPD development remais subject of discussion, the cigarette smoke is well recognized as a risk factor for the disease [3].

Chemokines such as CXCL1 and CXCL8 as well as cytokines TNF, IL-1β, IL-6, and IL-17 are chemotactic factors that attract inflammatory cells to the injured lung, principally neutrophils and monocyte-derived macrophage [4–7], where the pulmonary destruction initiates, compromising the alveolar parenchyma [8]. Exacerbated activity of metalloproteinases from neutrophils in COPD patients is responsible for destruction of alveolar parenchyma [9–12]. In COPD, neutrophils release proteinases into lung milieu, such as metalloproteases MMP-9 and MMP-12, result in emphysema [13] where the immune system switches to a Th17 response to promote the perpetuation of inflammation [14]. The effects of matrix metalloproteinase (MMP) can be inhibited by tissue inhibitors of metalloproteinase (TIMP) secreted by several cells [15]. During the pathogenesis of COPD, the balance between the effects of MMP and its TIMP is dysregulated [16–18], since that MMP released by neutrophils overlaps with TIMP activity with consequent pulmonary tissue destruction.

In parallel to the cytokine storm, the transcription factors NF-κB and the balance between STAT3/SOCS3 (suppressor of cytokine signaling 3) signaling are also present in the COPD pathogenesis through secretion of pro-inflammatory mediators, such as TNF, IL-8, IL-33, CXCL1, CXCL9, and CCL2 from bronchial epithelial cells [19, 20]. Some authors have evidenced an unbalanced SOCS3/STAT3 in *in vivo* COPD as well as in emphysematous patients [21–23]. This phenomenon is characterized by a reduced SOCS3 expression associated with increased STAT3 causing pulmonary fibrosis.

Cigarette pollutants can directly trigger pathogen-associated molecular patterns (PAMPs) such as toll-like receptors (TLRs), particularly TLR2 and TLR4, to initiate pattern recognition [24]. TLRs are present in dendritic cells, alveolar macrophages, neutrophils, and epithelial cells, and they have been correlated to lung inflammation caused by COPD [3]. Among them, the expression of TLR2, TLR4, and TLR9 is elevated in monocytes and TLRs are associated with number of sputum neutrophils, secretion of pro-inflammatory cytokines, and lung function impairment [25–27]. This is a reflex of the immune dysfunction observed in COPD [28, 29].

Some airways structural cells, such as the bronchial epithelium, when exposed to cigarette smoke secrete pro-inflammatory mediators activating alveolar macrophages as well as attracting neutrophils and activated lymphocytes towards the injured tissue [13, 30]. In fact, the airway epithelial cells are interface between innate and adaptive immunity. Moreover, the bronchial epithelial cells also discharge transforming growth factor-β (TGFβ), which triggers fibroblast proliferation for tissue remodeling [14, 31]. Therefore, small airway-wall remodeling strongly contributes to airflow limitation in COPD, decline in lung function, and poor responses to available therapies [32–34].

Due to the high morbidity and the limitations of existing COPD treatments [1, 35], innovative action is needed against airway inflammation as well as lung emphysema to better control the disease. One effective treatment for COPD may be to attenuate immune response driven to pro-inflammatory mediators and at the same time upregulate the secretion of anti-inflammatory proteins in lung milieu. Therefore, the ability of probiotics to modulate the immune response and the effects of their use to prevent the development of various chronic diseases, including COPD and asthma, has caught the attention of many researchers [36–40]. Little is known, however, concerning the nature of the probiotic-host cell interactions, or how these

interactions could be manipulated to obtain stronger regulatory responses in treatment against COPD.

Thus, we aim to investigate whether the oral feeding with probiotic *Lactobacillus rhamnosus* can beneficially modulate the immune response and attenuate lung inflammatory response in *in vivo* COPD model induced by cigarette smoke.

## 2. Material and methods

### 2.1. Animals

Three-month-old male C57Bl/6 mice were used. They were purchased from Center for the Development of Experimental Models (CEDEME) of the Federal University of São Paulo (UNIFESP), housed under controlled humidity, light and temperature conditions, inside ventilated polyethylene cages, in the vivarium located at Science and Technology Institute at the UNIFESP in São José dos Campos, SP, Brazil. The animals had food (Nuvilab–Quimtia. Brazil) and water ad libitum. The mice were anesthetized with ketamine (100 mg/kg) and xylazine (10 mg/kg) via i.p. and euthanized with excess anesthetics. The experiments were approved according to CONCEA (2016) and the Research Ethics Committee on Animal Use (CEUA) of UNIFESP under the register 9034130216.

### 2.2. Induction of COPD and preparation of the cigarette smoke extract (CSE)

The *in vivo* COPD was induced in C57Bl/6 mice by inhaling smoke from 14 cigarettes for 60 days, 7 days/week, for 30 min. The smoke was pumped into a plastic box measuring 42 cm (length) × 28 cm (width) × 27 cm (height), where the animals were kept and passively inhaled the cigarette smoke. For the *in vitro* experiments, the cigarette smoke extract (CSE) was prepared through the burning of 14 commercial cigarettes (tar: 13 mg; nicotine: 1.10 mg; carbon monoxide: 10 mg) using a vacuum machine (Nevoni– 1001 VF-PE. Series: 304—Brazil) with -11 Kpa to be incorporated into PBS.

### 2.3. Oral feeding with *Lactobacillus rhamnosus*

The mice were treated via gavage with probiotic *Lactobacillus rhamnosus* (*Lr*) ($1 \times 10^9$ CFU/0.2 mL PBS/mouse) (Liane Laboratory, Ribeirão Preto, SP) each day for seven days prior to the COPD induction and then 3 times/week until euthanasia. The experimental protocol is illustrated in Fig 1.

### 2.4. Murine bronchial epithelial cells and culture conditions

The lungs were removed and immersed in sterile enzymatic solution for digestion with dispase II for 60 min. After digestion, the cells were resuspended in cell basal medium that contained growth factors for epithelial cells and placed in petri dishes for 20 min. Adherent cells were collected and resuspended in RPMI 1640 plus fetal bovine serum, penicillin, and streptomycin and then maintained in culture until the third passage. The cells were isolated, plated and restimulated with 2.5% cigarette smoke extract (CSE) incorporated into the culture medium. The treatment with *Lactobacillus rhamnosus* (*Lr*) was performed 1 h before CSE addition in culture medium with murine bronchial epithelial cells. Then 24 hours after CSE addition, the culture supernatants were removed, and stored at -40˚C until use.

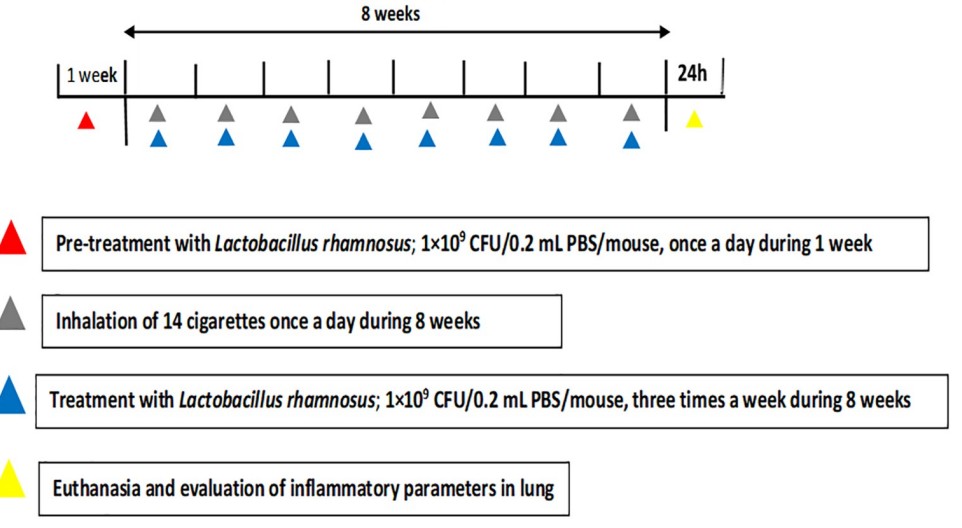

**Fig 1. Time schedule of COPD model and probiotic treatment.** Male C57Bl/6 mice were exposed to inhalation of cigarette smoke (14 cigarettes; 30 minutes/day; 7 days/week; during 60 days) for COPD induction and treated with *Lactobacillus rhamnosus (Lr)* ($10^9$ CFU/0.2 mL PBS/mouse) for seven days prior to the COPD induction and, after that, 3 times/week until the day of the euthanasia.

## 2.5. Human bronchial epithelial cells (BEAS-2B) and culture conditions

The lineage of human bronchial epithelium cells (BEAS-2B (ATCC® CRL-9609™)) were isolated from normal human bronchial epithelium obtained from autopsy of healthy individuals and were acquired from American Type Culture Collection (Manassas, VA). BEAS cells were cultured in small airway cell basal medium that contained growth factors for epithelial cells. The cells used were between the 45th and 55th generation passages. BEAS cells were washed with medium and introduced into each well of 24-well culture plates in triplicate at a concentration of $6\times10^5$ cells.mL$^{-1}$. After 12 hours, BEAS cells were exposed to cigarette smoke extract (CSE) and probiotic ($1\times10^5$ UFC of *Lr*). The CSE was made from 1 unfiltered cigarette which was burned in 10 mL of culture medium. A vacuum pump was used at a pressure of -11 Kpa so that the cigarette smoke could be incorporated into the culture medium. Cells were stimulated with 2.5% CSE incorporated into the culture medium. The treatment with probiotic was performed 1 h before CSE addition in culture medium with BEAS cells. Then, 24 hours after CSE addition, the culture supernatants were removed, and stored at -40°C until use.

## 2.6. Experimental groups

The animals were randomly divided into 3 groups of 7 animals each: control (pure air inhalation for 60 days); COPD (14 cigarettes smoke inhalation for 60 days; 30 min a day; 7 days/week); *Lr* + COPD (14 cigarettes smoke inhalation for 60 days; 30 min a day, 7 days/week + *Lactobacillus rhamnosus*). The animals received *Lr* ($10^9$ CFU/0.2 mL PBS/mouse) every day for the entire week before inhalation of CS and, after that, 3 days/week until euthanasia. For *in vitro* assays, murine bronchial epithelial cells or BEAS cells were divided in 3 groups: control (culture medium alone), COPD (murine bronchial epithelial cells or BEAS cells exposed to CSE), and *Lr* + COPD (murine bronchial epithelial cells or BEAS cells exposed to *Lr* 1-hour prior to addition of CSE in culture medium). The murine bronchial epithelial cells or BEAS cells were incubated with a density of $5\times10^5$ /mL of *Lr*.

## 2.7. Bronchoalveolar lavage fluid (BALF)

After the mice were euthanized with excess anesthetics, the trachea was cannulated, and lungs were rinsed with 0.5 mL of cold PBS (saline). This was followed by 2 additional washings with the same saline volume. Total and differential cell counts of BALF were determined by hemocytometer and cytospin preparation stained with Instant-Prov (Newprov, Brazil). Number of macrophages, neutrophils, and lymphocytes were scored by light microscopy.

## 2.8. Histology and image analysis

After the euthanasia, the lungs were carefully removed, perfused, and fixed with 4% paraformaldehyde for 24 h at a positive pressure (20 cm $H_2O$), for histological examination. Paraffin (Sigma-Aldrich Co., St. Louis, MO, USA) was used to embed the fixed tissue. Lung segments of approximately 5μm were stained with hematoxylin and eosin (Sigma-Aldrich Co.) for morphometric analysis of pulmonary emphysema. The parameters analyzed were peribronchial inflammation (polymorphonuclear cells), alveolar wall enlargement (Lm), deposition of collagen fibers, and destruction of elastic fibers. Five airways of all animals were imaged at 400 X magnifications using a Nikon Eclipse E-200 microscope camera and the software Image Pro-Plus 4.0.

**2.8.1. Alveolar enlargement.** Sections of lung tissue were stained with hematoxylin and eosin (Sigma-Aldrich Co.), and the increased air space was evaluated by the linear mean of the alveolar intercept (Lm) in twenty fields selected from each slide of lung tissue, with amplification of 200 X. The destruction of the alveolar septa was evaluated by the technique of counting points in twenty fields randomly throughout the pulmonary parenchyma (excluding fields presenting airways and pulmonary vessels).

**2.8.2. Peribronchial inflammation.** Peribronchial inflammation was obtained through analyzing the space between the basal membrane and adventitia. The number of polymorphonuclear cells were evaluated in this specific area. Polymorphonuclear cells were counted according to morphological criteria; groups were blinded. The results were expressed as number of cells per square millimeter. The number of polymorphonuclear cells per square millimeter of lung tissue was presented graphically.

**2.8.3. Collagen fibers.** The collagen deposition in the airways was performed with the addition of Picrosirius staining (Sigma-Aldrich Co.). The density of the collagen fibers was measured using a standardized color threshold (red) by the CellSens software from the region between the basal membrane of the epithelium to the adventitial airway. The results were expressed as $\mu m^2$ of collagen fibers/collagen per $\mu m^2$ of lung tissue area.

**2.8.4. Elastic fibers.** The destruction of elastic fibers in the airways was performed with the addition of Verhoeff Van Grieson (Sigma-Aldrich Co.) staining for elastic fiber marking. In brief, five airway tissues per animal (all animals of all groups) were subjected to image analysis using the CellSens software. The density of the elastic fibers was measured using a standardized color threshold (brown) by the CellSens software for the region between the basal membrane of the epithelium to the adventitial airway. The results were expressed in $\mu m^2$ of elastic fibers/elastic fibers per $\mu m^2$ of lung tissue area.

## 2.9. Cytokines and TGF-β in BALF, in bronchial epithelial cells and lung tissue

The levels of cytokine, chemokines and TGF-β in BALF, in murine bronchial epithelial cells, as well as in BEAS cells were assessed using ELISA kits for mice or human. The ELISA assay kit for mice was also employed to measure the SOCS3 concentration in lung tissue. All ELISA kits

were used in accordance with the manufacturer's instructions. Briefly, 200 μL of BALF were used in a quantitative sandwich enzyme immunoassay technique. For this, a mouse specific cytokine and chemokine monoclonal antibody was pre-coated on a microplate. Standards, control and colors are pipetted into wells and the concentration of cytokine present is activated by the immobilized antibody. After washing off non-activated chemicals, a polyclonal antibody binds to a specific enzyme to each cytokine and is then added to wells. After washing to remove any unbound antibody-enzyme reagent, a substrate solution is added to the wells. An enzymatic reaction produces a blue product that turns yellow when a stop solution is added. The intensity of the measurement is proportional to the amount of each cytokine evaluated in specific lysate kits in the initial step. The sample values are then read on the standard curve". Values are expressed as pg/mL deduced from standard runs in parallel with recombinant cytokines, chemokines and TGF-β.

## 2.10. Real-time Polymerase Chain Reaction (PCR) for MMP-9, MMP-12, TIMP1, TIMP2, TLR2, TLR4, TLR9, NF-κB, STAT3 and SOCS3

The mRNA expression in the animal lung tissue, in murine bronchial epithelial cells, as well as in BEAS cells was quantified by real-time PCR for MMP-9: `CGGATTTGGCCGTAT TGGGC` (forward) and `TGATGGCATGCACTGTGGTC` (reverse) and MMP-12: `TTTGACCCACTTCGCC` (forward) and `GTGACACGACGGAACAG` (reverse), TIMP-1: `CCACGAATCAAGAGACC` (forward) and `GGCCCGTGATGAGAAAC` (reverse) and TIMP-2: `GGTAGCCTGTGAATGTTCCT` (forward) and `ACGAAAATGCCCTCAGAAG` (reverse), TLR-2: `GAGCATCCGAATTGCATCACC` (forward) and `CCCAGAAGCATCACATGACAGAG` (reverse), TLR-4: `CATGGATCAGAAACT−CAGCAAAGTC` (forward) and `CATGCCATGCCTTGTCTTCA` (reverse), and TLR-9: `CAGC−TAAAGGCCCTGACCAA` (forward), and `CCACCGTCTTGAGAATGTTGTG` (reverse), plus the transcription factors NF-B: `CCGGGAGCCTCTAGTGAGAA` (forward) and `TCCATTTGTGAC−CAACTGAACGA` (reverse), STAT3: `TACCAGCCCTCCAATCAAAG` (forward) and `GGTCACA−CAGCACACAATCC`, and SOCS3: `CTGCAGGAGAGCGGATTCTACT` (forward) and `GCTGTC GCGGATAAGAAAGG` (reverse). The tests were conducted in accordance with the manufacturer's specifications. Briefly, 1μg of the total RNA was used for cDNA synthesis. Reverse transcription (RT) was performed in a 200μL solution in the presence of 50mM Tris-HCl (pH 8.3), 3mM MgCl$_2$, 10mM dithiothreitol, 0.5mM dNTPs, and 50ng random oligonucleotides with 200 units of reverse transcriptase (Invitrogen ™). Reaction conditions were: 20 ˚C for 10 min, 42 ˚C for 45 min, and 95 ˚C for 5 min. A 7000-sequence detection system (ABI Prism, Applied Biosystems®) was used through the SYBRGreen kit (Applied Biosystems®) and the values obtained normalized against the internal control gene GAPDH: `CGGATTTGGCCGTATTGGGC` (forward) and `TGATGGCATGCACTGTGGTC` (reverse).

## 2.11. Immunohistochemistry for NF-κB and STAT3 in lung tissue

For immunohistochemistry analysis, the paraffin-embedded sections of lung tissues were deparaffinized with xylene and then rehydrated. Section slides were incubated with 3% hydrogen peroxide for 10 min, then in 5% BSA in PBS blocking solution for 20 min, and after, incubated overnight with anti-NF-κB antibody (Cell signaling Technology) in blocking solution at 4 ˚C. After washing with PBS, the slides were treated with biotinylated secondary antibody for 20 min, streptavidin-HRP (horseradish peroxidase) for 20 min, and 3,3N-Diaminobenzidine Tetrahydrochloride for 10 min. The slides were then washed, and counter stained with hematoxylin. Slides were evaluated by microscopy, and the positive cells exhibited yellow or brown particles.

## 2.12. Statistics

The results were evaluated through the Analysis of Variance (ANOVA) and the Tukey-Kramer Multiple Comparison Test to determine the differences between the groups. The analysis were performed using Sigma Stat 3.1 software and graphs using GraphPad Prism 5.0 software. The results were considered significant when $p < 0.05$.

# 3. Results

## 3.1. *Lactobacillus rhamnosus* attenuate the cigarette-induced airway inflammation

The analysis of cellular content in BALF revealed that in response to cigarette smoke, the COPD group presented with an increase in the total number of cells in the airways, in comparison to the control, non-smoking group (Fig 2A). In concordance with the characteristic inflammatory response observed in COPD manifestation, the infiltrating cells were constituted by macrophages (2B), neutrophils (2C), and lymphocytes (2D). In contrast, it is possible to observe in the *Lr* group that probiotic feeding inhibited the influx of inflammatory cells into the airways. This phenomenon was accomplished by a significant attenuation in the levels of pro-inflammatory molecules (Fig 3). The exposure to cigarette smoke increased the levels of both pro-inflammatory cytokines, such as IL-1β (3A), IL-6 (3B), TNF-α (3C), KC (3D), IL-17 (3E), and TGF-β (3J) in BALF, in comparison to control group. Controversially, the *Lr* group presented with a significant reduction in the levels of both pro-inflammatory cytokines and chemokines when compared with the COPD mice. For notice, the inflammatory response

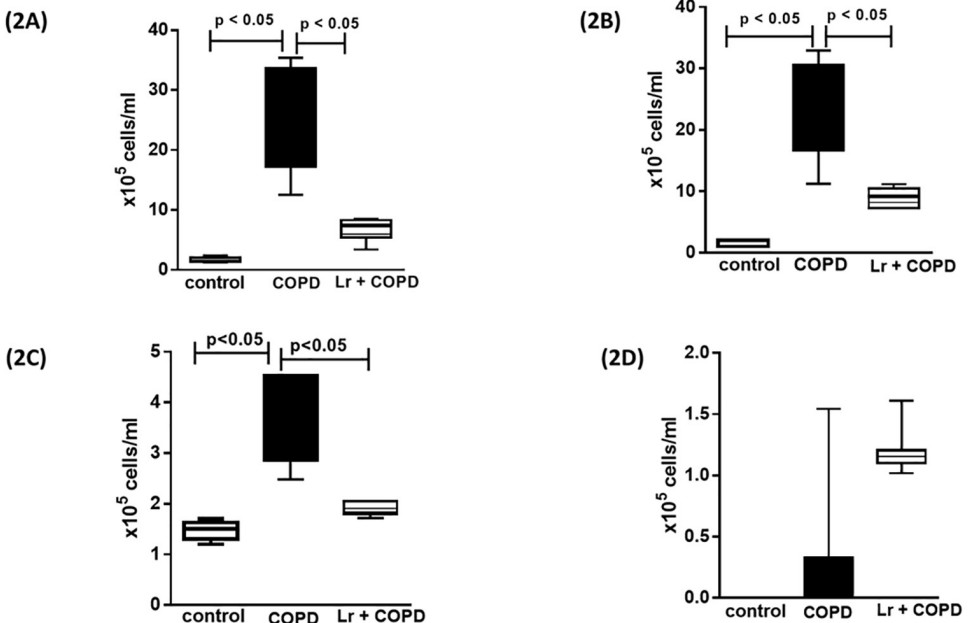

**Fig 2. Leukocyte lung infiltration.** After exposure of C57Bl/6 male mice to cigarette smoke and treatment with *Lactobacillus rhamnosus* (*Lr*), the total cells (2A) and inflammatory cells were counted (x10⁵) in BALF in millimeters by the morphometric evaluations of cytospin preparations. Pulmonary inflammation was represented by the influx of specific leukocytes; neutrophil (2B), macrophage (2C), and lymphocytes (2D) in BALF fluid. All cell counts were obtained from the control, COPD and *Lr* + COPD groups. No significant difference between *Lr* group and control group (data not shown). Each plot represents mean ± SEM from 7 different animals. The experiments were performed in triplicate. Results were considered significant when $p < 0.05$.

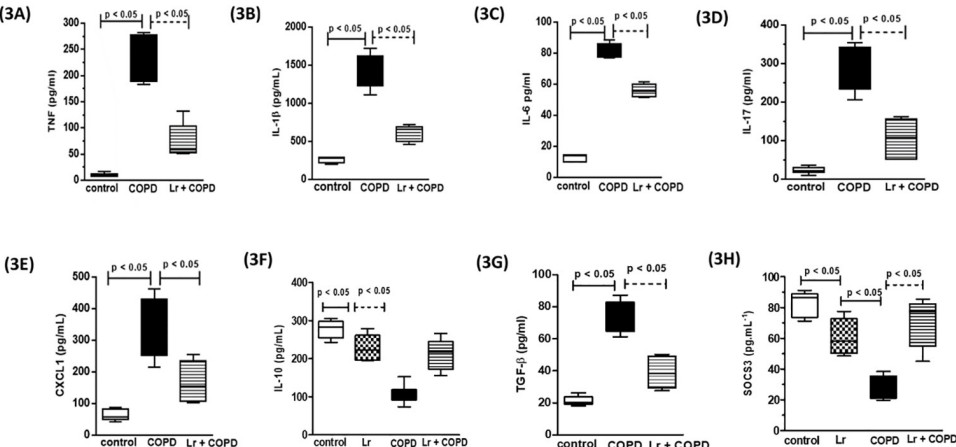

**Fig 3. Cytokines and chemokines in BALF and SOCS3 in lung tissue.** The BALF obtained from control, COPD, and *Lr* + COPD groups was prepared for analysis of pro- and anti-inflammatory cytokines. The mediator's TNF (3A), IL-1β (3B), IL-6 (3C), IL-17 (3D), CXCL1 (3E), IL-10 (3F), TGF-β (3G) in BALF and SOCS3 (3H) in lung tissue were assayed by enzyme-linked immunosorbent assay (ELISA). Each plot represents mean ± SEM from 7 different animals. The experiments were performed in triplicate. Results were considered significant when p < 0.05.

observed in the COPD group was correlated with a significant reduction in the levels of IL-10 (3K) in the airways, whereas in the *Lr* this anti-inflammatory cytokine is increased. For pro-inflammatory mediators, there was no difference between the *Lr* group and the control group (data not shown).

### 3.2. *Lactobacillus rhamnosus* attenuates pulmonary remodeling

In concordance with the findings observed in the BALF, the COPD group presented with an inflammatory response in lung tissue (Fig 4), with marked influx of polymorphonuclear cells into the parenchyma (4A). Also, these animals exhibited signals characteristic of tissue remodeling as alveolar wall enlargement (4B), and collagen deposition (4C) and elastic fibers destruction (4D). In contrast, it is possible to observe in the *Lr* group that probiotic feeding reduced the peribronchial inflammation (the influx of polymorphonuclear cells) as well as alveolar enlargment, collagen deposition and elastic fibers destruction. In morphometric studies, there was no significant difference between the *Lr* group and the control group (data not shown).

### 3.3. *Lactobacillus rhamnosus* modulates the balance between metalloproteases and tissue inhibitors of metalloproteases

In COPD, pulmonary remodeling is correlated with the deregulation in the balance between MMP and its inhibitors (TIMP). The analysis of lung tissue by quantitative PCR revealed that CS inhalation induced a significant increase in the mRNA expression of both MMP-9 (5A) and MMP-12 (5B) accomplished by inhibition in the expression of the genes associated with TIMP-1 and TIMP-2 proteins (5C and 5D, respectively). On the other hand, oral feeding with *Lr* sustained the expression of the mRNA for MMP-9 and MMP-12 in levels comparable to those found in control animals, and partially restored the expression of both TIMP-1 and TIMP-2 genes. No significant difference between *Lr* group and control group (data not shown) (Fig 5).

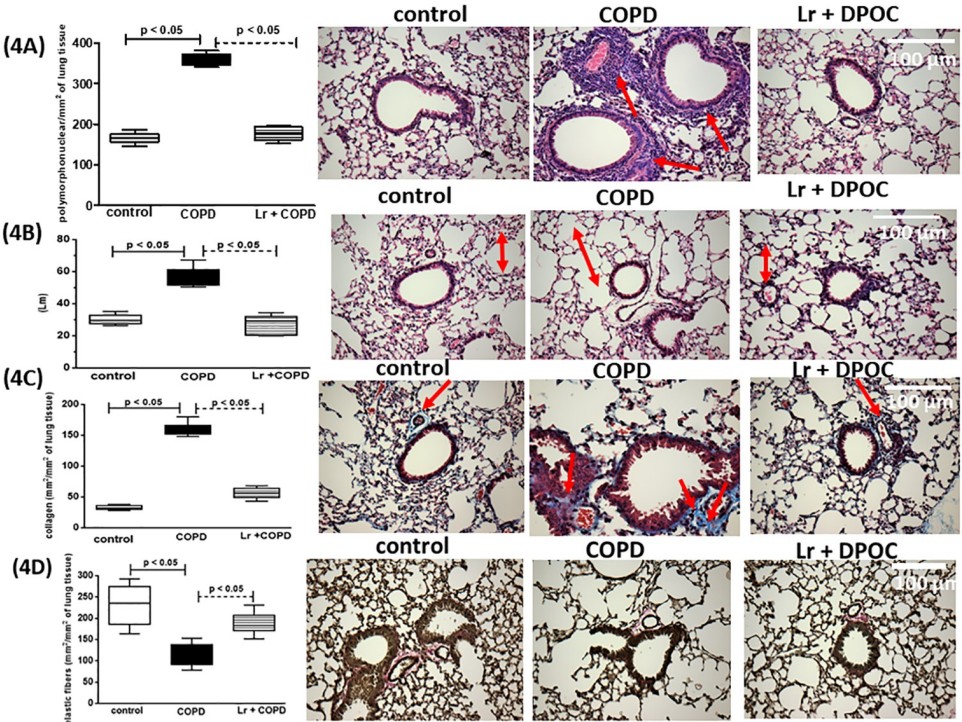

**Fig 4. Airway morphometry.** After exposure of C57Bl/6 male mice to cigarette smoke and treatment with *Lactobacillus rhamnosus* (*Lr*), sections (5 μm) of formalin-fixed lungs were stained with hematoxylin and eosin for histological examination in control, COPD, and *Lr* + COPD groups. (Original magnification, ×200). (4A) Quantification of polymorphonuclear cells in airway wall, (4B) Alveolar enlargement (Lm), (4C) Collagen fibers deposition, and (4D) Fiber elastics destruction were measured as described in Material and Methods section. Each plot represents mean ± SEM from 7 different animals. Results were considered significant when p < 0.05.

### 3.4. *Lactobacillus rhamnosus* downregulates expression of STAT3 and NF-κB in lung tissue

The activation of the STAT3 pathway and, consequently, the induction of NF- κB transcription factor regulate the expression of genes associated with inflammation in lung diseases and are correlated with disease severity. In concordance, inhalation of CS induced a significant increase in the mRNA expression of NF-κB and STAT3-related genes in lung tissue from COPD mice, in comparison to control group (Fig 6A and 6B, respectively). In contrast, the *Lr* group presented with a lower expression of these genes when compared to COPD mice, in levels similarly to those found in control group. These data were corroborated with immunohistochemical staining of lung tissue (Fig 6A and 6B). No significant difference between *Lr* group and control group (data not shown).

### 3.5. The beneficial effect of *Lactobacillus rhamnosus* is associated with a reduction in the expression of toll-like receptors in lungs

Because TLR engagement plays an important role in COPD pathogenesis, we decided to determine the status of mRNA expression to different TLRs in our model. Fig 7 demonstrates that among the TLR studied, the expression of mRNA for TLR2 (7A), TLR4 (7B), and TLR9 (7C) increased in cigarette smoke challenged-mice compared to control group. On the other hand, the treatment with *Lr* reduced the expression of induced a significant reduction in these genes

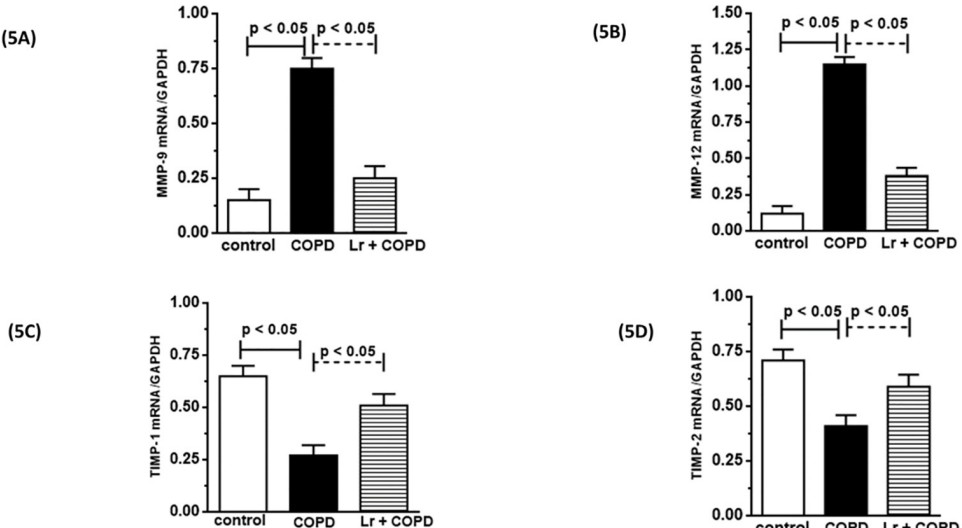

**Fig 5. Metalloproteases in lung tissue.** The mRNA expression of the MMP-9, MMP-12, TIMP-1 and TIMP-2 in lung from the control, COPD, and *Lr* + COPD groups is illustrated in Fig 5. After exposure of C57Bl/6 male mice to cigarette smoke and treatment with *Lactobacillus rhamnosus* (*Lr*), the mRNA expression for MMP-9 (5A), MMP-12 (5B), TIMP-1 (5C), and TIMP-2 (5D) in lung tissue were evaluated through Real Time-PCR. The values were normalized by the GAPDH expression and expressed by arbitrary units. Each bar represents mean ± SEM from 7 different animals. The experiments were performed in triplicate. Results were considered significant when $p < 0.05$.

in comparison with COPD group. No significant difference between *Lr* group and control group (data not shown).

### 3.6. *Lactobacillus rhamnosus* modulates the secretion of inflammatory mediators in murine bronchial epithelial cells

The airway epithelium is central to the pathogenesis of COPD. Therefore, we investigated the secretion of cytokines and chemokines from murine bronchial epithelial cells stimulated with CSE, treated with *Lr*, or stimulated with CSE and treated with *Lr*. As shown in Fig 8, the secretion of TNF (8A), IL-1β (8B), IL-6 (8C), CXCL1 (8D), from CSE-bathed murine bronchial epithelial cells increased compared to control group. On the contrary, the CSE-exposed murine bronchial epithelial cells secreted lower levels of IL-10 (8E), TGF-β (8F) as well as SOCS3 (8G) than murine bronchial epithelial cells from control group. The pre-incubation with *Lr* probiotic inhibited the secretion of all cytokines, with exception of SOCS3, TGFβ and IL-10, which were upregulated, even in comparison to control group. For pro-inflammatory mediators, there was no difference between the *Lr* group and the control group (data not shown).

### 3.7. *Lactobacillus rhamnosus* modulates the secretion of inflammatory mediators in human bronchial epithelial cells (BEAS)

We investigated the secretion of cytokines and chemokines from human bronchial epithelial cells (BEAS) stimulated with CSE or stimulated with CSE and treated with *Lr*. As shown in Fig 9, the secretion of TNF (9A), IL-1β (9B), IL-6 (9C), and CXCL8 (9D) from CSE-bathed BEAS cells increased compared to control group. On the other hand, CSE-exposed BEAS cells secreted lower levels of IL-10 (9E) as well as TGFβ (9F) compared to control group. The same effect was observed with levels of SOCS3 (9G) The presence of probiotic in BEAS cell culture

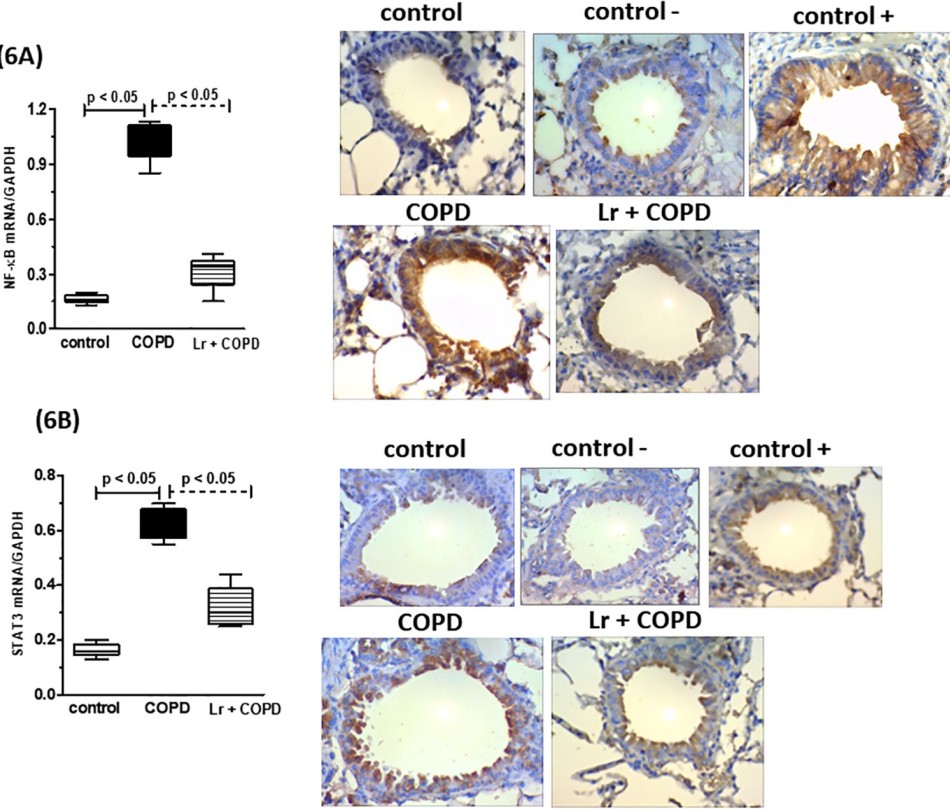

**Fig 6. Transcription factors in lung tissue.** The mRNA expression of the NF-κB and STAT3 in lung from the control, COPD, and *Lr* + COPD groups is illustrated. After exposure of C57Bl/6 male mice to cigarette smoke and treatment with *Lactobacillus rhamnosus* (*Lr*), the mRNA expression for NF-κB (6A) and STAT3 (6B) in lung tissue were evaluated through Real Time-PCR. The values were normalized by the GAPDH expression and expressed by arbitrary units. For immunohistochemical localization of NF-κB and STAT3 in lung tissue of mice from control, COPD and L*r* + COPD groups, the positive reaction was visualized as a yellowish-brown stain. Each plot represents mean ± SEM from 7 different animals. Results were considered significant when p < 0.05.

stimulated with CSE markedly inhibited the secretion of all cytokines investigated herein, with exception of SOCS3, TGFβ and IL-10, which were upregulated in comparison to control cells. For pro-inflammatory mediators, there was no difference between the *Lr* group and the control group (data not shown).

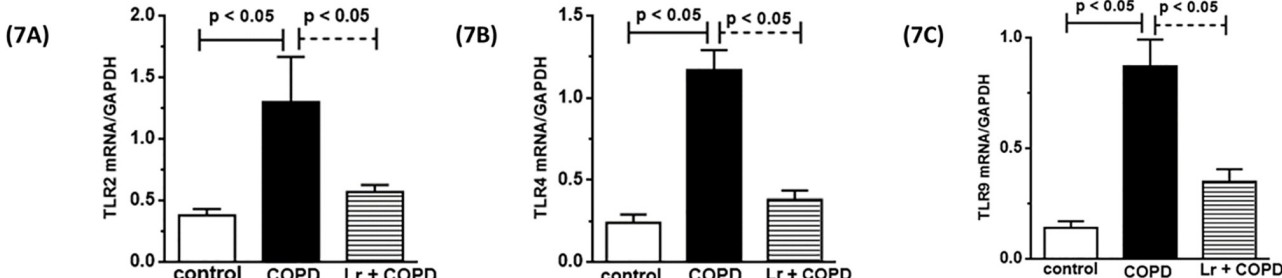

**Fig 7. Toll-like receptors in lung tissue.** After exposure of C57Bl/6 male mice to cigarette smoke and treatment with *Lactobacillus rhamnosus* (*Lr*), the mRNA expression for TLR2 (7A), TLR4 (7B) and TLR9 (7C) in lung tissue of control, COPD and Lr + COPD groups was evaluated through Real Time-PCR. The values were normalized by the GAPDH expression and expressed by arbitrary units. Each bar represents mean ± SEM from 7 different animals. The experiments were performed in triplicate. Results were considered significant when p < 0.05.

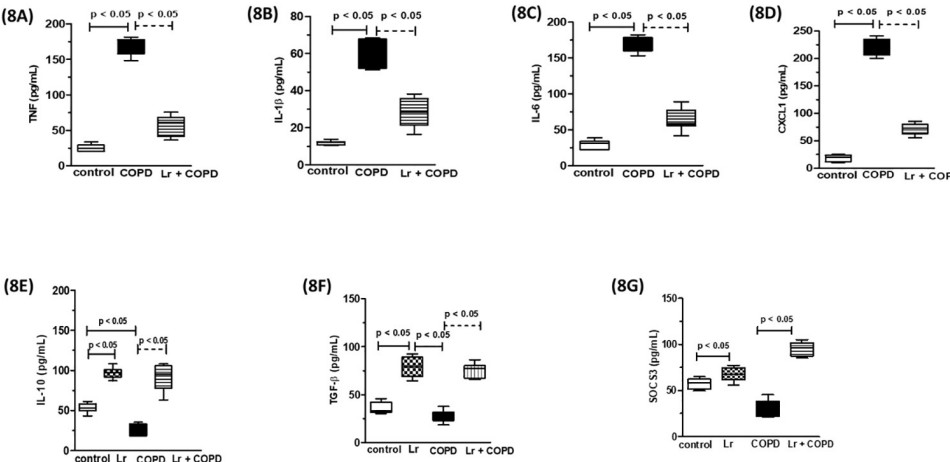

**Fig 8. Murine bronchial epithelial cells.** Cytokines and chemokines secretion from murine bronchial epithelial cells of control and COPD groups and treated with *Lr* 1 hour before addition of CSE is illustrated in Fig 8. The inflammatory mediator's TNF (8A), IL-1β (8B), IL-6 (8C), CXCL1 (8D), IL-10 (8E), TGF-β (8F) and SOCS3 (8G) in supernatant of murine bronchial epithelial cells were assayed by enzyme-linked immunosorbent assay (ELISA). The assays were performed in triplicate. Results were considered significant when p < 0.05.

## 3.8. *Lactobacillus rhamnosus* downregulates NF-κB, STAT3 and SOCS3 in murine bronchial epithelial cells and in BEAS cells

The *in vivo* findings indicated that the anti-inflammatory effects of *Lr* treatment was associated with the modulation of the genes associated with NFκB and STAT3 pathways (Fig 10). In concordance, both the murine bronchial epithelial cells (10A and 10B) and human bronchial epithelial cells (10C and 10D) cells presented with increased expression of mRNA to NF-κB and STAT3 genes upon exposure to CSE, in comparison to control cells. The *Lr* treatment maintained the expression of these genes at levels similar to those observed in the control cells,

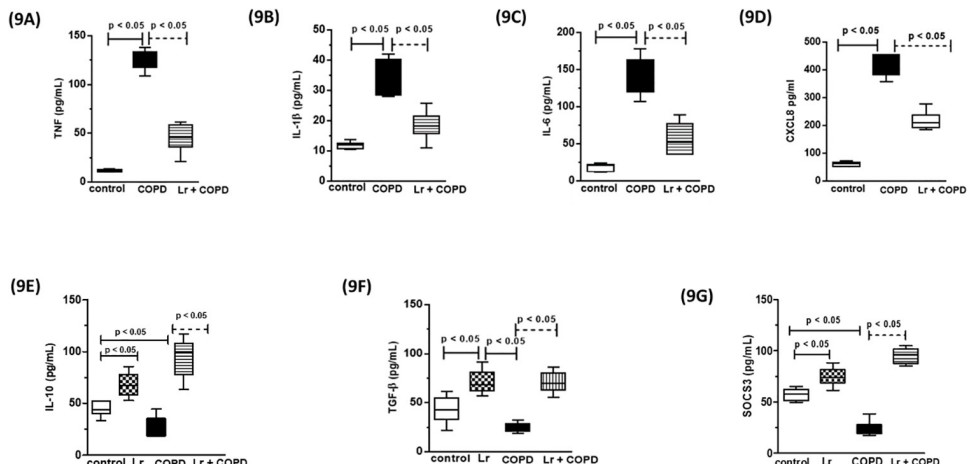

**Fig 9. Human bronchial epithelial cells (BEAS).** Cytokines and chemokines secretion from BEAS cells stimulated with CSE and treated with *Lr* are illustrated in Fig 9. The inflammatory mediator's TNF (9A), IL-1β (9B), IL-6 (9C), CXCL8 (9D), IL-10 (9E), TGFβ (9F) and SOCS3 (9G) in supernatant of human bronchial epithelial cells were assayed by enzyme-linked immunosorbent assay (ELISA). The assays were performed in triplicate. Results were considered significant when p < 0.05.

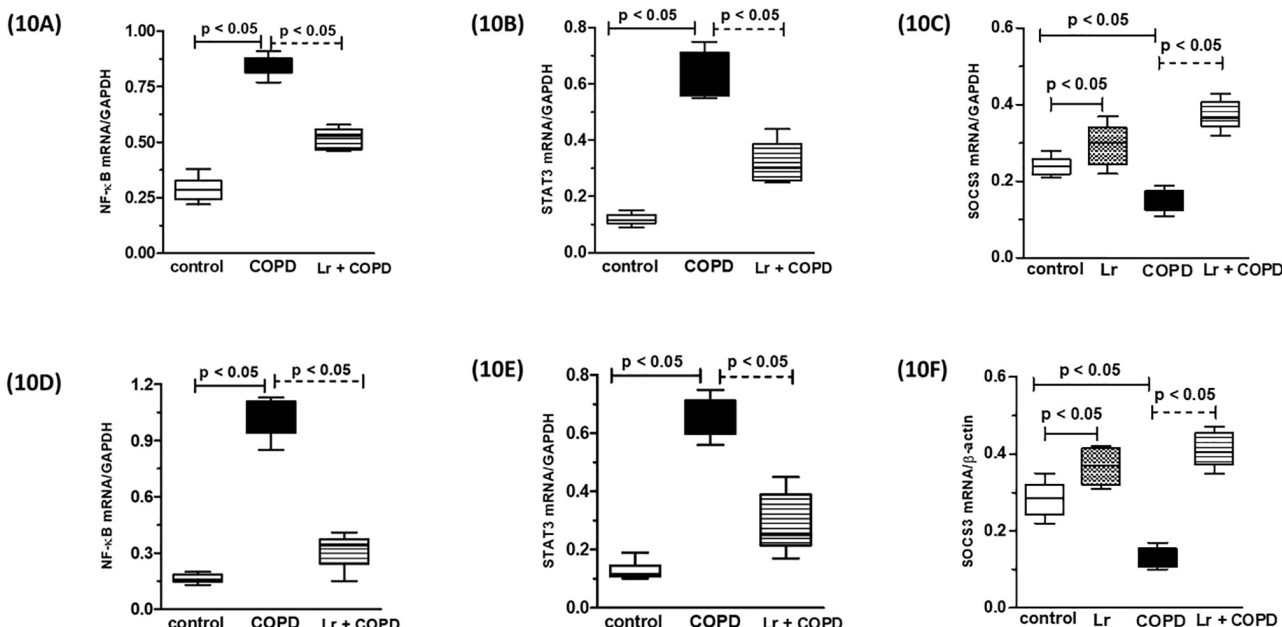

**Fig 10. Transcription factors in murine bronchial epithelial cells and in BEAS.** The mRNA expression of the NF-κB, STAT3 as well as SOCS3 in murine bronchial epithelial cells from the control, COPD, and *Lr* + COPD groups is illustrated in Figs 10A, 10B and 10C, respectively. Figs 10D, 10E and 10F presents the mRNA expression of the NF-κB, STAT3 as well as SOCS3 in human bronchial epithelial cells (BEAS) from the control, COPD, and *Lr* + COPD groups. After exposure of both airway epithelial cells, murine and human, to cigarette smoke extract (CSE) and treatment with *Lactobacillus rhamnosus* (*Lr*), the mRNA expression for NF-κB, STAT3 and SOCS3 in lung tissue was evaluated through Real Time-PCR. The values were normalized by the GAPDH expression and expressed by arbitrary units. The assays were performed in triplicate. Results were considered significant when $p < 0.05$.

corroborating the notion that the beneficial effects associated with the probiotic involves the control of the STAT3 pathway. On the contrary, both the murine bronchial epithelial cells (10C) and the BEAS cells (10F) stimulated with CSE presented a level lower of SOCS3 compared to control group. The oral feeding with *Lr* restored the SOCS3 levels in murine bronchial cells as well as in BEAS cells to values similar to control. STAT3 and NFκB pathways are regulated by molecules such as SOCS3, which lower levels has been associated with COPD. For pro-inflammatory transcription factors, there was no difference between the *Lr* group and the control group (data not shown).

## 4. Discussion

The present study demonstrates the ability of the probiotic *Lactobacillus rhamnosus (Lr)* to control lung inflammation in cigarette smoke (CS)-induced COPD experimental model. *Lr* feeding attenuated both the migration of inflammatory cells to the lung and tissue remodeling features, such as alveolar enlargement and exacerbated deposition of collagen and mucus secretion. Although the mechanisms involved in this phenomenon remain an object of study, the probiotic mitigated the cytokine storm associated with COPD pathogenesis, maintaining the equilibrium between transcription factors that regulate the production of pro and anti-inflammatory molecules.

Our findings corroborate previous studies showing that that CS inhalation induces a robust migration of inflammatory cells to the lung environment, mainly macrophage and neutrophils [41–43]. In response to CS, pulmonary cells produced pro-inflammatory cytokines such as IL-1β, IL-6 and TNF-α resulting in the secretion of several chemokines. The increase in CXCL1

levels promotes the migration and differentiation of monocytes in lung tissue, amplifying the inflammatory process [44–46]. The chronic inflammation of the lungs results in alterations in the parenchyma architecture, a process known as tissue remodeling, due an unbalance between active MMPs and its inhibitors, TIMP [47, 48]. In concordance with this notion, CS group presented, in association with pulmonary neutrophilia, alveolar enlargement as well as loss of alveolar parenchyma, collagen deposition, and destruction of elastic fibers. These structural alterations were accomplished by a significant increase in the mRNA expression for MMP-9 and MMP-12 that was inversely to gene expression for TIMP-1 and TIMP-2. Despite the cytokine/chemokine storm and the consequent cascade of events induced by CS inhalation, probiotic feeding attenuated the inflammatory process both in the airways space and lung parenchyma.

Although the mechanisms behind this effect are allusive, we found that *Lr* feeding increased both the levels of IL-10 in the BALF and SOCS3 in lung tissue, even when compared to control animals, indicating that probiotic induced an anti-inflammatory stead state. This idea is corroborated by the fact that *Lr* feeding sustained the levels of mRNA for TLRs, one of the major players in CS-induced COPD, and for pro-inflammatory transcription factors, such as NFκB and STAT3, at levels comparable to those from control animals. It is noteworthy that the group of mice that received only Lr (group Lr) was added in the Figures when there was significant difference from the Lr group compared to the control group. It happened in the concentration of IL-10, TGF-β and SOCS3 proteins, that is, proteins that guarantee an anti-inflammatory millieu in the lung. These results reinforce the idea that the insertion of certain probiotics in the diet can guarantee an anti-inflammatory balance in COPD. Our results also show that airway epithelial cells participate as target of Lr for secretion of the same anti-inflammatory mediators observed in in vivo COPD. These data strongly suggest that daily oral feeding protects the bronchial epithelial cells through increasing of anti-inflammatory response. This result is important because airway epithelial cells are the first barrier between the airways and the environment, and they are therefore activated by a variety of agents, including cigarette smoke.

Some authors have highlighted the complex roles of TLRs in the pathogenesis of COPD and suggested that the activation of TLR2 and/or inhibition of TLR4 may be novel therapeutic strategies for the treatment of COPD [49]. Despite it, most studies show that TLR4 stimulation followed by TLR2 stimulation does not cause tolerance but enhances cytokine production from activated neutrophils instead [50]. This may be a relevant mechanism by which bacteria cause excessive inflammation in COPD patients. In addition, some authors demonstrated that CSE-exposed human bronchial epithelial cells secrete CXCL8 and IL-1β via TLR4 [51]. Although presenting cytosol-initiated signaling, unlike membrane receptor TLR2 and TLR4, the TLR9 has been reported to be important for lung neutrophils infiltration in COPD [52]. Indeed, these authors show that TLR9 agonists induce neutrophil migration in mice with COPD, and that TLR9 knockout mice do not show neutrophil migration after exposure to cigarette smoke. Within the airways of smokers, the TLR9 is the crucial signaling for the pro-inflammatory mediators secretion that contribute to the accumulation of neutrophils [53, 54]. Based on our results, it seems possible that Lr has the ability to control lung inflammation by reducing TLRs receptor expression regardless of cell signaling and localization of each investigated TLR.

Because epithelial cells are the interface between innate and adaptative immunity and a growing body of evidence supports a major role for non-immune pulmonary cells in COPD [55–57], we decided to study the response of epithelial cells to probiotic stimulation. The exposure of murine or human epithelial cells to cigarette extract (CSE) resulted in the secretion of several pro-inflammatory cytokines and chemokines and induced expression of mRNA to

STAT3 and NFκB, mirroring the observed in the *in vivo* model. In contrast, probiotic stimulation made the epithelial cells refractory to the inflammatory provocation provided by CE.

The proposal of action mechanism for *Lr* effect in which the bronchial epithelium is the important target was confirmed when the probiotic modulated the secretion of pro- and anti-inflammatory cytokines in human bronchial epithelial cells stimulated with CSE. Moreover, our results show that the anti-inflammatory effect of *Lr* on cytokines secretion from CSE-exposed human bronchial epithelial cells was due to the downregulation of both transcription factors, NF-κB and STAT3. The modulation of transcription factors SOCS3 and STAT3 by probiotic was also observed in human bronchial epithelial cells. This reinforces the idea that the *in vivo* action mechanism of *Lr* involves the signaling pathway NF-κB/STAT3/SOCS3 in human bronchial epithelium cells in order to attenuate both the lung inflammation and the exacerbation of immune response in lung microenvironment.

Finally, several strains of *Lactobacillus*, as well as its structural components, and microbial-produced metabolites can stimulate epithelial cell signaling pathways which can prevent cytokine and oxidant-induced epithelial damage thereby promoting cell survival through increased production of cytoprotective molecules [57]. Our results demonstrated that *Lr*, by itself, increased secretion of both the IL-10 and the TGF-β secretion as well as SOCS3 levels in human bronchial epithelial cells unstimulated with CSE, which supports the idea that daily supplementation with probiotic may protect the lung milieu through of airway epithelial cells, since the IL-10 can suppress pro-inflammatory genes and the TGF-β can guarantee the integrity of airway epithelial barrier.

In conclusion, the present manuscript describes by the first time that *Lr* modulates the secretion of pro- and anti-inflammatory molecules from human airway epithelial cells through of restoring the equilibrium between the transcription factors NF-kB/STAT-3 and SOCS3, and it seems to be an important action mechanism of probiotic in order to control lung inflammation as well as airway remodeling in COPD.

## Author Contributions

**Conceptualization:** F. Aimbire.

**Data curation:** J. L. Carvalho.

**Investigation:** F. Aimbire.

**Methodology:** J. L. Carvalho, M. Miranda, A. K. Fialho, H. Castro-Faria-Neto, E. Anatriello, A. C. Keller.

**Project administration:** F. Aimbire.

**Writing – original draft:** J. L. Carvalho.

**Writing – review & editing:** A. C. Keller, F. Aimbire.

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
