## [Decision Letter · Decision Letter 0]

9 Dec 2019

PONE-D-19-30819

Oral feeding with probiotic *Lactobacillus rhamnosus* attenuates cigarette smoke-induced COPD in C57Bl/6 mice: Relevance to inflammatory markers in human bronchial epithelial cells

PLOS ONE

Dear Dr. Aimbire,

Thank you for submitting your manuscript to PLOS ONE. After careful consideration, we feel that it has merit but does not fully meet PLOS ONE’s publication criteria as it currently stands. Therefore, we invite you to submit a revised version of the manuscript that addresses the points raised during the review process.

We would appreciate receiving your revised manuscript by Jan 23 2020 11:59PM. To enhance the reproducibility of your results, we recommend that if applicable you deposit your laboratory protocols in protocols.io, where a protocol can be assigned its own identifier (DOI) such that it can be cited independently in the future. For instructions see: http://journals.plos.org/plosone/s/submission-guidelines#loc-laboratory-protocols

We look forward to receiving your revised manuscript.

Kind regards,

Hong Wei Chu

Academic Editor

PLOS ONE

Journal Requirements:

**When submitting your revision, we need you to address these additional requirements:**

**Please ensure that your manuscript meets PLOS ONE's style requirements, including those for file naming. The PLOS ONE style templates can be found at http://www.plosone.org/attachments/PLOSOne_formatting_sample_main_body.pdf and http://www.plosone.org/attachments/PLOSOne_formatting_sample_title_authors_affiliations.pdf**Please ensure that you refer to Figure 5 in your text as, if accepted, production will need this reference to link the reader to the figure.

Reviewers' comments:

Reviewer's Responses to Questions

**Comments to the Author**

1. Is the manuscript technically sound, and do the data support the conclusions?

Reviewer #1: Partly

2. Has the statistical analysis been performed appropriately and rigorously? 

Reviewer #1: Yes

3. Have the authors made all data underlying the findings in their manuscript fully available?

Reviewer #1: Yes

4. Is the manuscript presented in an intelligible fashion and written in standard English?

Reviewer #1: Yes

5. Review Comments to the Author

Reviewer #1: The pathogenesis of COPD is one that still needs to be fully elucidated in order for better treatment for patients suffering from this disease. The manuscript takes an interesting approach to utilize probiotics in order to prevent onset of COPD in a mouse model as well as pro-inflammatory effects of CSE on epithelial cells. The authors, intriguingly, show a positive effect with the addition of Lactobacillus rhamnosus (Lr) in the lungs of a COPD mouse model and epithelial cells from mice and humans. This manuscript, however, needs major revisions in many aspects. First, the order of the figures is not easy for the reader to follow the line of reasoning. For example, it is the opinion of this reviewer, that figure 4 be moved to figure 2 to emphasize the mouse model is an appropriate model for COPD. Secondly, the mouse model is a 60-day long experiment, yet at the end, mRNA for cytokines and chemokines is measured over protein concentration. After such a chronic exposure, protein levels would be more impactful than mRNA. Third, the BALF data, although very interesting, does not seem to fit within this manuscript. Since Lr is being fed to the mice, the bacteria would therefore would have a greater impact on endothelial cells within the lung more than epithelial or BAL. There is also a possibility that Lr is influencing proteins within the liver and having an effect on the liver/lung axis more than within the lung tissue itself. Without data looking at liver proteins and/or endothelial cell responses in the lung, the BALF data seems to be out of place. Lastly, the authors make assumptions about the timeline of experiments and concentrations of stimuli. For example, it was never described why the concentration of Lr was determined for this paper. Has this been shown previously? The experimental design for the smoking studies also seems quite aggressive with 14 cigarettes smoked in ½ hour. Overall, this paper could be rewritten with stronger data to emphasize the importance of probiotics on COPD pathogenesis.

Other comments:

• It was unclear how BALF was prepared for ELISA cytokine/chemokine expression. Were the cells isolated and re-stimulated? Were they isolated, seeded on a plate, and left for 24 hours post isolation for cytokine expression?

• The H&E staining in Figure 4 does not emphasize specific cell type within the lungs.

• All pictures are missing scale bar.

• Collage and elastin fibers are very hard to see in Figure 4. If proximity to the airway is not emphasized within the text, it would be helpful to have a picture closer of the collage and elastin.

• Figure 6: Is there a positive and negative control for these stains? It seems as though entire cells are showing positive of NFkB especially within the epithelial cell layer. This could be a place where the authors can tie in the epithelial data easily by doing IF staining for NFkB and an epithelial marker, thereby further strengthening their decision to further investigate epithelial cells.

• Figure 7: The relevance of selective TLR mRNA is not discussed. While it was mentioned that differences in TLR expression has been shown in COPD, this is still controversial based on the patient. The reason for investigating these 3 TLRs was also not mentioned and considering all three have different intracellular signaling, this was a confusing choice.

• For the epithelial studies – the mouse data can be moved to supplemental has it follows the human data. It would also be helpful to either include Lr alone as a treatment within graphs or do not have it. It hard to follow why it was included in some graphs, but not others.

6. PLOS authors have the option to publish the peer review history of their article (what does this mean?). If published, this will include your full peer review and any attached files.

Reviewer #1: No

---

## [Author Response · Author response to Decision Letter 0]

21 Jan 2020

PLOS ONE

Responses to Reviewer

PONE-D-19-30819

“Oral feeding with probiotic Lactobacillus rhamnosus attenuates cigarette smoke-induced COPD in C57Bl/6 mice: Relevance to inflammatory markers in human bronchial epithelial cells” 

Review Comments to the Author

Reviewer #1

1. The pathogenesis of COPD is one that still needs to be fully elucidated in order for better treatment for patients suffering from this disease. The manuscript takes an interesting approach to utilize probiotics in order to prevent onset of COPD in a mouse model as well as pro-inflammatory effects of CSE on epithelial cells. The authors, intriguingly, show a positive effect with the addition of Lactobacillus rhamnosus (Lr) in the lungs of a COPD mouse model and epithelial cells from mice and humans. 

R: Dear Reviewer, although this is the first study that investigate the effect of Lr on inflammatory parameters that define the COPD, there are some authors that have shown the beneficial effects of Lr in the treatment of patients with allergic asthma, and these effects have been reproduced in murine asthma model. In addition, some authors have reported that Lr is able to influence immune and inflammatory cells, as well as, structural cells such as airway epithelial cells and lung fibroblast. For this reason, it is reasonable to suggest that Lr would also be able to attenuate lung inflammation in an in vivo COPD model induced by exposure to cigarette smoke. Finally, the results of the present manuscript are original and it does not seem intriguing, since a set of published data (see references) support our hypothesis. There is a mechanism of action described by Forsythe (2011) (it is in our references) for probiotics that suggest that the signaling of the anti-inflammatory effect of Lr is linked to cells of the pulmonary immune system. However, before of the present study, it was never described in in vivo COPD model.

2. This manuscript, however, needs major revisions in many aspects. First, the order of the figures is not easy for the reader to follow the line of reasoning. For example, it is the opinion of this reviewer, that figure 4 be moved to figure 2 to emphasize the mouse model is an appropriate model for COPD. 

R: The authors understand the Reviewer's point of view, but we would like to keep the order of the results presented in this manuscript as it represents the sequence of events in the experimental COPD model. As it may be noted, the present study begins by assessing inflammatory parameters in vivo, and based on these results we investigated the cellular mechanisms involved in COPD. Lastly, the in vivo results led us to investigate the airway epithelial cells. Therefore, we kindly urge the Reviewer to maintain the sequence in the manuscript due to the explanation below:

- initially the cells reach the lung (BALF cellularity with predominance of neutrophils)

- these cells secrete cytokines (cytokines in BALF)

- the cytokines in turn alter lung tissue (histology and morphometry)

- the COPD-related proteases contribute to tissue damage (MMP-9 and MMP-12)

- the transcription factors responsible for the synthesis of inflammatory mediators are present (NF-κB, STAT3)

- the receptors expression of immune response are also involved in the lung (TLR2, 4 and 9)

- the effect of Lr on in vivo COPD may be the result of its modulating effect on inflammatory mediators and transcription factors in bronchial epithelial cells.

3. Secondly, the mouse model is a 60-day long experiment, yet at the end, mRNA for cytokines and chemokines is measured over protein concentration. After such a chronic exposure, protein levels would be more impactful than mRNA. 

R: Dear Reviewer, in the present study, cytokine and chemokine levels were not assessed by mRNA. Instead, all cytokines and chemokines were evaluated by ELISA kits that indicate protein concentration of these inflammatory mediators in BALF as well as in lung tissue.

4. Third, the BALF data, although very interesting, does not seem to fit within this manuscript. Since Lr is being fed to the mice, the bacteria would therefore would have a greater impact on endothelial cells within the lung more than epithelial or BAL. There is also a possibility that Lr is influencing proteins within the liver and having an effect on the liver/lung axis more than within the lung tissue itself. Without data looking at liver proteins and/or endothelial cell responses in the lung, the BALF data seems to be out of place. 

R: Dear Reviewer, we find your remark very pertinent, however, this manuscript addresses the effect of Lr on lung inflammation observed in cigarette smoke exposure-induced COPD, so the characteristics of this type of lung inflammation are targets of Lr. One of the main features of lung inflammation is the large number of inflammatory cells that migrate to the lung. These cells can be detected in BALF, and are responsible for the early phase and progression of COPD because they secrete cytokines capable of attracting more and more inflammatory cells, perpetuating lung inflammation. This progression of inflammatory cell-induced inflammation in BALF culminates in irreversible changes in lung tissue, such as mucus deposition, increased collagen secretion, alveolar enlargment and destruction of elastic fibers. Thus, the evaluation of cellularity and cytokines secretion in BALF is an important marker in the treatment of COPD. Regarding to endothelial cells, it is important to highlights that the choice to analyze airway epithelial cells is not due to the effect of Lr, but due to the fact that airway epithelial cells orchestrate the initial response of the inflammatory process after contact with the toxic components present in cigarette smoke, since they secrete cytokines capable of acting as chemoattractants for neutrophils. Neutrophils secrete proteases that can destroy the lung parenchyma. The airway epithelial cells are the first line of defense against allergens and toxic components. In fact, the airway epithelial cells are the interface between innate and adaptive immunity in COPD. Thus, the Reviewer is right in suggesting the possibility that Lr may influence lung endothelial cells, however our focus was to evaluate the effect of Lr on airway epithelial cells due to its crucial importance for COPD. The Reviewer is also right to suggest that Lr may influence liver proteins, and that there may be a lung-liver axis. However, liver proteins alteration is not a marker of lung inflammation in COPD. In COPD, inflammatory mediators, such as cytokines and chemokines, secreted in the pulmonary environment are the most important inflammatory markers and predictors of disease severity and for this reason the lung is the target organ of our study. Moreover, the present manuscript shows results that are based on the description of the action mechanism of probiotics in lung inflammation published by Forsyhte, 2011 (see reference). In this study the author describes that the beneficial effect of probiotics, including Lr, is due to the initial recognition of non-pathogenic bacteria by antigen presenting cells in the gut-associated lymphoid tissue (GALT), from that moment on, these cells stimulate the lymphocytes that migrate to the lung and stimulate bronchial lymphoid tissue (BALT) to secrete anti-inflammatory mediators such as IL-10. Thus, although the liver-lung axis can be relevant because the liver participates in the metabolization of probiotics, the present study deals with the use of Lr in lung inflammation in COPD, and therefore, lung cells.

5. Lastly, the authors make assumptions about the timeline of experiments and concentrations of stimuli. For example, it was never described why the concentration of Lr was determined for this paper. Has this been shown previously? The experimental design for the smoking studies also seems quite aggressive with 14 cigarettes smoked in ½ hour. Overall, this paper could be rewritten with stronger data to emphasize the importance of probiotics on COPD pathogenesis.

R: Dear Reviewer, we have recently published a manuscript in “Cellular Immunology” journal entitled “Oral feeding of Lactobacillus bulgaricus N45.10 inhibits the lung inflammation and airway remodeling in murine allergic asthma: Relevance to the Th1/Th2 cytokines and STAT6/T-bet. Anatriello E, Cunha M, Nogueira J, Carvalho JL, Sá AK, Miranda M, Castro-Faria-Neto H, Keller AC, Aimbire F. Cell Immunol. 2019 341:103928. doi: 10.1016/j.cellimm.2019.103928. Based on that study, we decided to use the same concentration of Lr in the present manuscript. Concerning the in vivo COPD model, our study is adapted from the article published in Plos One in which mice were exposed to cigarette smoke for 75 days. “Human Tubal-Derived Mesenchymal Stromal Cells Associated with Low Level Laser Therapy Significantly Reduces Cigarette Smoke-Induced COPD in C57BL/6 mice”. Peron JP, de Brito AA, Pelatti M, Brandão WN, Vitoretti LB, Greiffo FR, da Silveira EC, Oliveira-Junior MC, Maluf M, Evangelista L, Halpern S, Nisenbaum MG, Perin P, Czeresnia CE, Câmara NO, Aimbire F, de Paula Vieira R, Zatz M, de Oliveira AP. PLoS One. 2015 25;10(9):e0139294. doi: 10.1371/journal.pone.0139294. Moreover, the animal induction protocol was approved by the Ethics Committee on Animal Study, as described in the “Material and Methods” section. Although the reviewer suggested that stronger results should be included to evidence the effect of Lr, we would like to point out that the effect of Lr was evaluated on the inflammatory parameters that characterize the COPD, from BALF cellularity to changes in lung architeture. In addition, our results show that human bronchial epithelial cells are targeted by Lr for controlling the lung inflammatory response in COPD. These results have clinical importance, and in this sense, this study makes a translational bridge the beneficial effect of Lr in murine bronchial epithelium to the human lineage BEAS. In addition, the present manuscript describes a mechanism of action for the anti-inflammatory and immunomodulatory effect of probiotic Lr. 

Other Comments:

1 • It was unclear how BALF was prepared for ELISA cytokine/chemokine expression. Were the cells isolated and re-stimulated? Were they isolated, seeded on a plate, and left for 24 hours post isolation for cytokine expression?

R: BALF - The bronchoalveolar lavage fluid (BALF) was obtained from three lung washes with phosphate buffered saline (PBS) (3 x 0.5 mL PBS). BALF was collected and centrifuged at 3000 rpm for 15 minutes at 4ºC and the supernatant collected and resuspended in 1ml PBS. The material was then stored in a freezer at -80ºC for analysis of cytokines through ELISA kit (DuoSet® Kit, R&D Systems) following the manufacturers' recommendations. BALF was prepared for ELISA as follow: “Briefly, 200 µL of BALF were used in a quantitative sandwich enzyme immunoassay technique. For this, a mouse specific cytokine and chemokine monoclonal antibody was pre-coated on a microplate. Standards, control and colors are pipetted into wells and the concentration of cytokine present is activated by the immobilized antibody. After washing off non-activated chemicals, a polyclonal antibody binds to a specific enzyme to each cytokine and is then added to the wells. After washing to remove any unbound antibody-enzyme reagent, a substrate solution is added to the wells. An enzymatic reaction produces a blue product that turns yellow when a stop solution is added. The intensity of the measurement is proportional to the amount of each cytokine evaluated in specific lysate kits in the initial step. The sample values are then read on the standard curve”. It is noteworthy that the description of how biological fluid samples (cellular supernatant, serum, sputum, BALF) or tissue homogenates are prepared for the ELISA can be found in the kit description in detail. Therefore, it is not common to describe the entire sample preparation steps in the "Material and Methods". 

Cells - The cells were isolated, plated and restimulated with 2.5% cigarette smoke extract (CSE) incorporated into the culture medium. Then, 24 hours after CSE addition, the culture supernatants were removed, and stored at -40°C until use for cytokine analysis through Elisa kits. It was inserted in subsection 2.4 in the “Material and Methods” section.

2 • The H&E staining in Figure 4 does not emphasize specific cell type within the lungs.

R: Dear Reviewer, the H&E staining pattern only illustrates the presence of inflammatory infiltrate in the peribronchial region. However, the technique of tissue morphometry allows to evaluate the specific type of inflammatory cell. For this reason, in the present study it was possible to evaluate the number of neutrophils in the peribronchial region. 

3 • All pictures are missing scale bar.

R: Dear Reviewer, the scale bars were inserted in the pictures as required.

4 • Collagen and elastin fibers are very hard to see in Figure 4. If proximity to the airway is not emphasized within the text, it would be helpful to have a picture closer of the collage and elastin.

R: Dear Reviewer, the figure related to collagen received “red arrows”, indicating the collagen clearly marked in blue. Thus, Figure 4C shows a bluish coloring in the COPD group and a reduction in blue color in the mice of the COPD group treated with Lr. In addition, we added “red arrows” in peribronchial inflammatory infiltrate, alveolar enlargment, and collagen deposition. The elastic fibers of the lung tissue were marked in a brownish tone. Therefore it is possible to visualize the whole Figure 4D with brownish color illustrating the elastic fibers.

5 • Figure 6: Is there a positive and negative control for these stains? It seems as though entire cells are showing positive of NFkB especially within the epithelial cell layer. This could be a place where the authors can tie in the epithelial data easily by doing IF staining for NFkB and an epithelial marker, thereby further strengthening their decision to further investigate epithelial cells.

R: Dear Reviewer, the positive and negative controls for stains (NF-κB and STAT3) were inserted in Figure 6. We agree with the Reviewer's suggestion about to localize via IF the NF-kB and STAT3 in bronchial epithelial cells, however we would like to point out that the choice to study bronchial epithelial cells is not related to the in vivo experiment where in fact the most intense staining seems to be located in the bronchial epithelial region. The choice to evaluate bronchial epithelial cells is due to their importance in the development and progression of COPD. Thus, even if there no participation of transcription factors NF-κB and STAT3 in lung tissue of COPD mice, both murine bronchial epithelial cells and BEAS cells would be investigated. 

6 • Figure 7: The relevance of selective TLR mRNA is not discussed. While it was mentioned that differences in TLR expression has been shown in COPD, this is still controversial based on the patient. The reason for investigating these 3 TLRs was also not mentioned and considering all three have different intracellular signaling, this was a confusing choice.

R: Dear Reviewer, it was inserted in the “Discussion” section. We would like to reaffirm that our goal was to investigate whether Lr could influence the lung expression of TLR2, 4, and 9 receptors reported by some authors as the most important innate immunity receptors in the pathogenesis of COPD. Thus, the localization and different cellular signaling of each TLR is not necessarily relevant, in principle, to understand whether Lr can attenuate the expression of these receptors in the lung. Thus, the choice to investigate TLR 2, 4, and 9 as targets of Lr in in vivo COPD was relevant. 

7 • For the epithelial studies – the mouse data can be moved to supplemental has it follows the human data. It would also be helpful to either include Lr alone as a treatment within graphs or do not have it. It hard to follow why it was included in some graphs, but not others. 

R: The authors would like to appeal to the Reviewer to understand that data on the effect of Lr on in vivo COPD is partly a reflex of the Lr effect on bronchial epithelial cells. Thus, the relevance of the present study is that these cells tha are crucial for the development and chronicity of COPD are targets of Lr. That way, if this data is moved to the supplemental data, the article relevance is lost and even title will also have to be changed. Thus, we would like to request that the results of Lr effect on murine and BEAS be grouped together with the main results. For the Lr group, results with Lr alone were not included in some experiments because there was no difference between the control group and the Lr group. Otherwise, when we evaluated the effect of Lr on anti-inflammatory mediators such as IL-10, TGF-beta and SOCS3, we noticed that there was a significant increase in the levels of these mediators in the Lr group that was not exposed to smoke. This effect was observed in BALF, murine bronchial epithelial cells and in BEAS cells. For this reason, the Lr group has been inserted only in these figures. The authors would also like to emphasize how interesting these results are because they show that Lr can create an anti-inflammatory environment even in mice and BEAS cells that have not been exposed to smoke.

---

## [Editor Report · Decision Letter 1]

18 Feb 2020

PONE-D-19-30819R1

Oral feeding with probiotic *Lactobacillus rhamnosus* attenuates cigarette smoke-induced COPD in C57Bl/6 mice: Relevance to inflammatory markers in human bronchial epithelial cells

PLOS ONE

Dear Dr. Aimbire,

Thank you for submitting your manuscript to PLOS ONE. After careful consideration, we feel that it has merit but does not fully meet PLOS ONE’s publication criteria as it currently stands. Therefore, we invite you to submit a revised version of the manuscript that addresses the points raised during the review process.

In Figure 4, neutrophil count data based on H&E staining is not convincing. There was no indication of neutrophils in the provided histology pictures. Immunostaining of neutrophils could be considered.

We would appreciate receiving your revised manuscript by 3/17/2020. To enhance the reproducibility of your results, we recommend that if applicable you deposit your laboratory protocols in protocols.io, where a protocol can be assigned its own identifier (DOI) such that it can be cited independently in the future. For instructions see: http://journals.plos.org/plosone/s/submission-guidelines#loc-laboratory-protocols

We look forward to receiving your revised manuscript.

Kind regards,

Hong Wei Chu

Academic Editor

PLOS ONE

---

## [Author Response · Author response to Decision Letter 1]

17 Mar 2020

PLOS ONE

Responses to Reviewer

PONE-D-19-30819R1

 “Oral feeding with probiotic Lactobacillus rhamnosus attenuates cigarette smoke-induced COPD in C57Bl/6 mice: Relevance to inflammatory markers in human bronchial epithelial cells” 

Review Comments to the Author

Reviewer #1

1. In Figure 4, neutrophil count data based on H&E staining is not convincing. There was no indication of neutrophils in the provided histology pictures. Immunostaining of neutrophils could be considered.

R: Dear Reviewer, we also take the opportunity to notify You that we made a mistake in Figure 4A. The correct caption of the graph is “polymorphonuclear/mm2 of lung tissue”. In this way, we also corrected the text exchanging “neutrophils” in the lung parenchyma for "polymorphonuclear cells". It is worth noting that this change does not compromise the characteristics of the in vivo COPD model, nor the interpretation nor the conclusions of results found in the present manuscript. Finally, the Reviewer is correct when affirm that to define the number of neutrophils in the lung tissue more accurately, a specific marker is needed. However, we would like to respectfully attach some manuscripts published in selective editorial policy journals where the authors evaluate the number of neutrophils in peribronchial region using the H&E staining pattern only. The manuscripts were inserted below:

- Lanças T, Kasahara DI, Prado CM, Tibério IF, Martins MA, Dolhnikoff M. Comparison of early and late responses to antigen of sensitized guinea pig parenchymal lung strips. J Appl Physiol (1985). 2006 100(5):1610-6. 

- Rigonato-Oliveira NC, Mackenzie B, Bachi ALL, Oliveira-Junior MC, Santos-Dias A, Brandao-Rangel MAR, Delle H, Costa-Guimaraes T, Damaceno-Rodrigues NR, Dulley NR, Benetti MA, Malfitano C, de Angelis C, Albertini R, Oliveira APL, Abbasi A, Northoff H, Vieira RP. Aerobic exercise inhibits acute lung injury: from mouse to human evidence Exercise reduced lung injury markers in mouse and in cells. Exerc Immunol Rev. 2018; 24:36-44.

- Davino-Chiovatto JE, Oliveira-Junior MC, MacKenzie B, Santos-Dias A, Almeida-Oliveira AR, Aquino-Junior JCJ, Brito AA, Rigonato-Oliveira NC, Damaceno-Rodrigues NR, Oliveira APL, Silva AP, Consolim-Colombo FM, Aimbire F, Castro-Faria-Neto HC, Vieira RP. Montelukast, Leukotriene Inhibitor, Reduces LPS-Induced Acute Lung Inflammation and Human Neutrophil Activation. Arch Bronconeumol. 2019 55(11): 573-580.

---

## [Editor Report · Decision Letter 2]

6 Apr 2020

Oral feeding with probiotic *Lactobacillus rhamnosus* attenuates cigarette smoke-induced COPD in C57Bl/6 mice: Relevance to inflammatory markers in human bronchial epithelial cells

PONE-D-19-30819R2

Dear Dr. Aimbire,

We are pleased to inform you that your manuscript has been judged scientifically suitable for publication and will be formally accepted for publication once it complies with all outstanding technical requirements.

With kind regards,

Hong Wei Chu

Academic Editor

PLOS ONE
---

## [Editor Report · Acceptance letter]

13 Apr 2020

PONE-D-19-30819R2 

Oral feeding with probiotic *Lactobacillus rhamnosus* attenuates cigarette smoke-induced COPD in C57Bl/6 mice: Relevance to inflammatory markers in human bronchial epithelial cells 

Dear Dr. Aimbire:

I am pleased to inform you that your manuscript has been deemed suitable for publication in PLOS ONE. Congratulations! Your manuscript is now with our production department. 

With kind regards,

on behalf of

Dr. Hong Wei Chu 

Academic Editor

PLOS ONE